# C5aR2 Regulates STING-Mediated Interferon Beta Production in Human Macrophages

**DOI:** 10.3390/cells12232707

**Published:** 2023-11-25

**Authors:** Oliver Wright, Anna Harris, Van Dien Nguyen, You Zhou, Maxim Durand, Abbie Jayyaratnam, Darren Gormley, Luke A. J. O’Neill, Kathy Triantafilou, Eva Maria Nichols, Lee M. Booty

**Affiliations:** 1Immunology Network, GSK, Stevenage SG1 2NY, UK; 2School of Biochemistry and Immunology, Trinity College Dublin, D02 VR66 Dublin, Ireland; 3Systems Immunity Research Institute, Cardiff University, Cardiff CF14 4XW, UK; 4Division of Infection and Immunity, School of Medicine, Cardiff University, Cardiff CF14 4XW, UK; 5Immunology Research Unit, GSK, Stevenage SG1 2NY, UKeva-maria.x.nichols@gsk.com (E.M.N.)

**Keywords:** complement, C5a, anaphylatoxin receptor, C5aR2, pattern recognition receptor, innate immune sensing, cGAS-STING, interferon beta

## Abstract

The complement system mediates diverse regulatory immunological functions. C5aR2, an enigmatic receptor for anaphylatoxin C5a, has been shown to modulate PRR-dependent pro-inflammatory cytokine secretion in human macrophages. However, the specific downstream targets and underlying molecular mechanisms are less clear. In this study, CRISPR-Cas9 was used to generate macrophage models lacking C5aR2, which were used to probe the role of C5aR2 in the context of PRR stimulation. cGAS and STING-induced IFN-β secretion was significantly increased in C5aR2 KO THP-1 cells and C5aR2-edited primary human monocyte-derived macrophages, and STING and IRF3 expression were increased, albeit not significantly, in C5aR2 KO cell lines implicating C5aR2 as a regulator of the IFN-β response to cGAS-STING pathway activation. Transcriptomic analysis by RNAseq revealed that nucleic acid sensing and antiviral signalling pathways were significantly up-regulated in C5aR2 KO THP-1 cells. Altogether, these data suggest a link between C5aR2 and nucleic acid sensing in human macrophages. With further characterisation, this relationship may yield therapeutic options in interferon-related pathologies.

## 1. Introduction

The complement system is a complex network of proteins which initiate and modulate the immune response. Complement system pattern recognition receptors (PRR) initiate a tightly regulated proteolytic cascade, which generates effector protein fragments with opsonic, pro-inflammatory and pore-forming functions. The complement system thereby plays an important role in immunity by directing and regulating the innate and adaptive immune responses [1], and offers therapeutic targets for inflammatory and degenerative diseases [2].

C5a, a key effector of the complement system, mediates a range of effects on immune cells through its receptors complement component 5a receptor 1 (C5aR1) and complement component 5a receptor 2 (C5aR2). C5aR1 is a 7-transmembrane G protein-coupled receptor (GPCR), which binds C5a [3] and signals via heterotrimeric G proteins to activate various pro-inflammatory signalling pathways including mitogen-activated protein (MAP) kinase and nuclear factor-kappa B (NF-κB) [4,5]. This results in an array of immunomodulatory effects including the regulation of pro-inflammatory cytokine secretion, as well as the chemotaxis and degranulation of neutrophils [6]. C5aR2 is a non-canonical anaphylatoxin receptor of the complement system. It is predominantly expressed in the blood and spleen [7] by myeloid cells [8], and is also a 7-transmembrane receptor. However, C5aR2 lacks three critical GPCR-associated motifs [9]. As a result, C5aR2 does not function as a canonical GPCR as it cannot bind G proteins or elicit cAMP or Ca^2+^ ion mobilisation [10]. In comparison to C5aR1, the function of C5aR2 is poorly understood. Early work identified C5aR2 as a decoy receptor acting as a C5a sink [11]. However, more recent studies have demonstrated that C5aR2 is able to signal independently and modulate the immune response [9,12,13].

Given the recent discovery of the independent functions of C5aR2 in immune signalling, we sought to further characterise its role in the context of myeloid immunity. Previous studies have shown interactions of C5aR2 with β-arrestins [10,14,15], as well as negative regulation of C5aR1-induced ERK1/2 phosphorylation [14] and PRR-induced cytokine secretion [16] in human monocyte-derived macrophages. Specifically, C5aR2 agonism using peptide agonists suppressed Toll-like receptor (TLR) 3, TLR4, TLR7, Dectin-2, Mincle and stimulation of interferon gene (STING)-induced cytokine secretion. These studies suggest independent and immunomodulatory functions for C5aR2. To further our understanding of C5aR2, and given the potential implications of C5aR2 in disease [9,12], we aimed to utilise genetic editing as a tool to provide further insight into the function of C5aR2 as a regulator of inflammatory signalling in myeloid cells.

Here, CRISPR-Cas9 was used to generate C5aR2 knockout (KO) THP-1 cells to enable the study of C5aR2 function in a monoclonal genetically null macrophage-like cell system in the context of PRR stimulation and cytokine production. An enhanced interferon (IFN) signature in C5aR2 KO THP1 cells was confirmed at transcript and protein levels, and in physiologically relevant human monocyte-derived macrophages CRISPR edited for C5aR2. Bulk transcriptomics identified differential regulation of IFN-based signalling nodes, including cytosolic DNA sensing and JAK-STAT signalling, further characterising the immunomodulatory role of C5aR2. This work elucidates a preliminary link between C5aR2 and STING signalling, with potential for implicating C5aR2 in interferon-related pathologies. Further molecular characterisation is required to fully understand the relevance and importance of this interaction and whether this could translate to a novel therapeutic opportunity.

## 2. Materials and Methods

### 2.1. Materials

THP-1 cells were sourced from ATCC (TIB-202). CRISPR-Cas9 consumables were obtained from Synthego (Redwood, CA, USA), Integrated DNA Technologies and Lonza. G3-YSD, H-151, BX795, LPS (from *E. coli* K12, Ultrapure), cAIM(PS)_2_ Difluor (Rp/Sp) and Pam3CSK4 were sourced from Invivogen. Human C5a (non-recombinant) was sourced from Complement Technologies. Peggy Sue consumables were sourced from BioTechne. Antibodies were sourced from Thermo Fisher Scientific (Waltham, MA, USA), Cell Signaling Technologies (Danvers, MA, USA) and Abcam (Cambridge, UK). Primary human CD14^+^ peripheral blood mononuclear cells were sourced from leukopaks supplied by Discovery Life Science (Huntsville, AL, USA) or BioIVT (Westbury, NY, USA) with suitable genetic sequencing consents in place. The human biological samples were sourced ethically, and their research use was in accordance with the terms of the informed consents under an IRB/EC approved protocol.

Culture medium was generated by combining RPMI 1640 medium (Gibco (Waltham, MA, USA)), 10% FBS (Gibco), 2 mM L-glutamine (Gibco) and 100 U/mL Penicillin-Streptomycin (Gibco).

### 2.2. THP-1 Cell Culture

Cryopreserved THP-1 cells were thawed at 37 °C, re-suspended in culture medium, then transferred to a 50 mL conical tube containing 10 mL pre-warmed culture medium for centrifugation at 300× *g* for 5 min. Supernatant was removed and cells were re-suspended in culture medium. Cells were initially cultured at 0.2 × 10^6^ cells/mL and passaged at 1 × 10^6^ cells/mL. Cells were cultured at 37 °C, 5% CO_2_, in a humidified environment. THP-1 cells were differentiated by plating at an appropriate density in culture medium + 20 nM phorbol 12-myristate 13-acetate (PMA, Merck, Darmstad, Germany) for 24 h, then medium was removed and replaced with culture medium (without PMA) for a further 24 h. THP-1 cells were used at passage 15 or below.

### 2.3. Primary Human Cell Isolation and Culture

Peripheral blood mononuclear cells were isolated from leukopaks (one per donor, five in total) from healthy human donors using density gradient centrifugation and CD14^+^ cells were isolated using an EasySep Human CD14 Positive Selection Kit II (STEMCELL Technologies, Vancouver, BC, Canada). CD14^+^ peripheral blood mononuclear cells were cultured in 75 cm^2^ flasks in culture medium + 100 ng/mL M-CSF (R&D Systems, Minneapolis, MN, USA) for 6 days. At day 6, primary human monocyte-derived macrophages (hMDMs) were detached using Detachin (AMSBIO, Oxford, UK) and used for CRISPR-Cas9 editing. Cells were then plated at 100,000 cells/well in 96-well plates and incubated at 37 °C for a further 10 days, with medium changes at day 3 and 6.

### 2.4. CRISPR-Cas9 Editing of THP-1 Cells and Primary Human Macrophages

Ribonucleoprotein (RNP) containing sgRNA and Cas9 was generated using published protocols [17,18]. In brief, 1.5 nmol sgRNA was reconstituted to 100 µM in IDTE buffer and diluted to 25 µM in Duplex Buffer (IDT, Leuven, Belgium). 1 µL 10 mg/mL Cas9 (IDT) was added, mixed and incubated at RT for 10 min to form RNPs. Electroporation and cationic lipid transfection were used to deliver RNP to THP-1 cells using P3 Primary Cell 4D Nucleofector Kit S (Lonza, Basel, Switzerland). 250,000 cells were suspended in 16.1 μL P3 Buffer (Lonza) per reaction, then mixed with 3.9 μL RNP and electroporated using protocol DE148 on a 4D-Nucleofector (Lonza).

Cells were incubated at 37 °C for 5 min, then transferred into 2 mL pre-warmed culture medium and incubated at 37 °C for 48 h. A sample of the polyclonal population was harvested for sequencing. After sequencing confirmed a satisfactory editing efficiency, monoclonal cell lines were generated using limiting dilution. Cells were counted, plated in 96-well plates at 1 cell/well (in 100 µL volume), and incubated at 37 °C until sufficient cell proliferation had occurred, which was monitored by observing medium colour changes and cell presence by light microscopy. Samples from individual wells were removed for sequencing, indels were confirmed by sequencing analysis, and selected monoclonal populations were cultured until sufficient cells were available for cryopreservation and experimental use. C5aR1 KO clones B6, C3 and G8, and C5aR2 KO clones D3, F3 and F7 were selected due to positive sequencing data and growth rate comparable to wild-type.

Primary hMDMs were cultured as above, before detachment from culture flasks using Detachin. Cells were counted, followed by electroporation and cationic lipid transfection using the protocol above.

### 2.5. Guide Sequences

The guides were designed using tools from Integrated DNA Technologies. The below table describes the guide sequence for each gene (Table 1).

### 2.6. PCR and Sanger Sequencing

PCR primers were designed to bind 150 bp up- and down-stream from PAM sites using Primer-BLAST (NCBI [19]). Cells were lysed using 50 µL lysis buffer (1 mM CaCl_2_, 3 mM MgCl_2_, 10 mM TRIS-EDTA 1% Triton X-100, 0.2 mg/mL Proteinase K) per 50,000 cells, then incubated at 65 °C for 10 min followed by 95 °C for 15 min in a thermocycler. 10 µL Phusion HF PCR Master Mix (Thermo Fisher), 7 µL H_2_O, 1 µL 10 µM forward primer and 1 µL 10 µM reverse primer were combined with lysate as template DNA, and PCR was performed using a SimpliAmp Thermal Cycler (Applied Biosystems, Waltham, MA, USA) with the following protocol: 98 °C 30 s, (98 °C 10 s, 65 °C 15 s, 72 °C 15 s) × 35, 72 °C 60 s, 4 °C. PCR product was visualised using agarose gel electrophoresis, purified using a QIAquick PCR Purification Kit (Qiagen, Hilden, Germany), and submitted to GENEWIZ for Sanger sequencing using the PCR primers as sequencing primers. Sequencing data was analysed using BioEdit 7.2.5 [20], Clustal Omega (Version 1.2.0) [21] and ICE Analysis 3.0 (Synthego, CA, USA).

### 2.7. Primer Sequences

Primers were designed using Primer Blast and sequences are described below in Table 2. 

### 2.8. Flow Cytometry

Cells were detached using TrypLE Express Enzyme (Gibco), washed in Cell Staining Buffer (BioLegend, San Diego, CA, USA) three times then re-suspended in Cell Staining Buffer and plated in 96-well plates at 250,000 cells/well. Cells were centrifuged at 350× *g* for 5 min, then incubated with a 1:20 dilution of Human TruStain FcX (BioLegend) at 4 °C for 10 min. Cells were stained using anti-C5aR1-APC (clone S5/1) (BioLegend), anti-C5aR2-APC (clone 1D9-M12) (BioLegend) or isotype control antibody-APC (IgG2a, κ) (BioLegend). Antibodies were used at 0.625 µg/mL in a staining volume of 200 µL for 250,000 cells. For extracellular staining, cells were incubated with staining antibodies at 4 °C for 20 min. For intracellular staining, cells were washed in Cell Staining Buffer three times, then re-suspended in 100 µL Fixation Buffer (BioLegend) and incubated at 4 °C for 5 min. Cells were washed in Cell Staining Buffer three times and permeabilised by incubating in 100 µL Intracellular Staining Permeabilisation Wash Buffer (BioLegend) at 4 °C for 5 min. Cells were washed in Cell Staining Buffer three times, then incubated with staining antibodies at 4 °C for 20 min.

Following staining, cells were washed three times and re-suspended in Cell Staining Buffer. Attune Performance Tracking Beads (Invitrogen, Waltham, MA, USA) were used to perform cytometer setup, and flow cytometry was performed using an Attune NxT (Life Technologies, Carlsbad, CA, USA). Data were collected using Attune NxT Software 3.1 (Life Technologies) and analysed using FlowJo 10.8.1.

### 2.9. Cytokine Detection

Supernatants were assessed for cytokine by U-Plex Assay (MSD; IL-1β, IL-6, IL-10, TNFα), Luminex (Thermo Fisher; IL-1β, IL-6, IL-10, TNFα, IL-18, GM-CSF) or Human IFN-β DuoSet ELISA (R&D Systems; IFN-β) according to the manufacturer’s instructions. ELISA data were acquired using a PHERAstar FSX (BMG Labtech, Ortenberg, Germany) and PHERAstar FSX 5.70 software (BMG Labtech).

### 2.10. Viability Assay

Following supernatant harvest, cell viability was assessed using Cell Titre-Glo Luminescent Cell Viability Assay (Promega, Fitchburg, WI, USA) according to the manufacturer’s instructions. Data were acquired using a PHERAstar FSX (BMG Labtech) and PHERAstar FSX 5.70 software (BMG Labtech).

### 2.11. Protein Detection by Peggy Sue

Cell lysates were generated by addition of Lysis Buffer (Bio-Techne) containing 1X HALT Protease and Phosphatase Inhibitor Cocktail (Thermo Fisher Scientific) for 15 min on ice. Lysates were centrifuged at 18,000× *g* for 20 min at 4 °C, and supernatants were transferred to pre-cooled microcentrifuge tubes.

Expression of STING (PA5-23381, Thermo Fisher Scientific), phospho-STING (Ser366; 50907, Cell Signaling Technology, MA, USA), IRF3 (ab68481, Abcam), phospho-IRF3 (Ser386; 37829, Cell Signaling Technology) and loading control β-actin (ab6276, Abcam) were assessed using a 12–230 kDa Separation Module and a Peggy Sue Automated Western Blot System (Bio-Techne) according to the manufacturer’s instructions. Briefly, samples were added to master mix containing internal standards and DTT and heated to 95 °C for 5 min. Samples were loaded into the plate, and capillary electrophoresis was performed using stacking and separation matrices, blocking solution (Antibody Diluent), primary antibody (as defined), biotin-conjugated secondary antibody (anti-rabbit), Streptavidin-HRP and peroxide/luminol-S mix for chemiluminescent signal detection. Compass for Simple Western 5.0.1 (Bio-Techne) was used to generate lane view plots, and peak area data were plotted as a ratio compared to the loading control. Uncropped images are available in this manuscript.

### 2.12. Fluorescent Microscopy

Cells cultured in Nunc Lab-Tek II Chamber Slide Systems (Thermo Fisher Scientific) were washed twice in PBS, then fixed by adding 200 µL 10% formalin solution for 10 min at RT. Cells were then washed and blocked in staining medium (PBS + 0.02% BSA) for 1 h at RT. Mouse anti-human C5aR1 mAb (Clone S5/1) (BioLegend) was diluted 1:100 into permeabilisation medium (staining medium + 0.02% saponin), and 200 µL was added to each chamber. Chambers were sealed and incubated at RT overnight. Cells were then washed three times using staining medium, before incubation with 1:100 dilution of Alexa Fluor 488 goat anti-mouse IgG (H + L) (Invitrogen) in permeabilisation medium at RT for 1 h in the dark. Cells were then washed three times using staining medium and incubated with 1 µg/mL Hoechst (Thermo Fisher Scientific) at RT for 15 min in the dark. Cells were washed three times in staining medium, chambers were removed from the slides, then Vectashield antifade mounting medium (Vector Laboratories) and cover slips were added to the slide and the slide was sealed. 3 representative images were acquired from each chamber using an Airyscan LSM880 (Zeiss, Oberkochen, Germany) with 405 nm and 488 nm lasers and a 20× oil immersion lens. Images were analysed using Zen 2.6 (Zeiss).

### 2.13. RNASeq and Analysis

PMA-differentiated THP-1 cells (WT, C5aR1 KO or C5aR2 KO) in 75 cm^2^ flasks were stimulated with culture medium containing 5 µg/mL cAIM(PS)_2_ Difluor (Rp/Sp), 50 ng/mL C5a or vehicle for 6 h. Supernatants were harvested for IFN-β ELISA to confirm response to stimulation and cells washed twice with PBS, then harvested using TrypLE Express Enzyme. Cells were centrifuged at 300× *g* for 5 min, the supernatant was removed and the cells were snap-frozen on dry ice. RNASeq was performed by GENEWIZ, comprising total RNA isolation, library preparation with poly(A) selection and 150 bp paired-end sequencing using an HiSeq 2500 (Illumina, San Diego, CA, USA). Before performing analyses, genes were pre-filtered, and only genes that were expressed in at least 50% of the samples were retained. To visualise similarity between samples, principal component analysis (PCA) was performed using variance-stabilising transformed data. For differentially expressed gene (DEG) analysis, DESeq2 was used for comparisons between groups of interest [22]. Potential batch effect was removed by fitting the design as “~Batch + condition”. Additionally, *p* values were adjusted using the Benjamini–Hochberg method for significance assessment. To perform pathway analysis, the gene lists were first pre-ranked and subsequently used as an input for fast gene set enrichment analysis (fgsea) using fgsea packages [23,24]. Pathways of interest were further visualized. All analyses and visualisation were generated using R 4.0.3 [25]. An adjusted *p* value of 0.05 was used as a cut-off for statistical significance. Further analysis was performed using STRING (Version 12.0) [26] and PANTHER 17.0 [27].

### 2.14. Statistics

Data were analysed using R 4.0.3 and Excel v2202 (Microsoft, Redmond, WA, USA). Figures were generated and statistical analysis was performed using Prism 8.1.2 (Graphpad, San Diego, CA, USA) and Spotfire 11.4.4 (Tibco, Palo Alto, CA, USA). Normality of datasets was assessed using Kolmogorov-Smirnov and Shapiro–Wilk tests. Statistical tests and replicate numbers are indicated in figure legends associated with each figure. *p* values are indicated as follows: ns: *p* > 0.05; *: *p* ≤ 0.05; **: *p* ≤ 0.01; ***: *p* ≤ 0.001; ****: *p* ≤ 0.0001.

## 3. Results

### 3.1. Generation of Monoclonal C5aR2 KO THP-1 Cells

An established high editing-efficiency CRISPR-Cas9 pipeline was used to generate monoclonal C5aR2 KO THP-1 cell lines (Figure 1A) [18]. The expression of C5aR2 in wild-type THP-1 cells was initially assessed by flow cytometry (Appendix A). C5aR2 was intracellularly expressed with a slight reduction in expression upon PMA differentiation. C5aR2 was not expressed extracellularly under these conditions. Having detected C5aR2 in wild-type cells, RNPs against exon 2 of C5aR2 were delivered to wild-type THP-1 cells and, after 48 h, cells sequenced to confirm a polyclonal population of C5aR2-edited cells. After limiting dilution, monoclonal cultures were screened for single-origin edits by Sanger sequencing (Appendix A) and monoclonal colonies confirmed for C5aR2 KO by flow cytometry (Figure 1B,C). Together, this confirmed the generation of monoclonal C5aR2 KO THP-1 cell lines for use in this study.

### 3.2. Assessing the Response of C5aR2 KO THP-1 Cells to PRR Stimulation

The role of C5aR2 in regulation of the innate immune response has been previously explored using C5a-based peptide agonists [14,16]. In this study, we assessed the relevance of C5aR2 in PRR signalling in a macrophage model. PMA-differentiated wild-type and C5aR2 KO THP-1 cells were stimulated with a range of PRR agonists, including TLR1/2 activation with Pam3CSK4, TLR4 activation with LPS and STING activation with cAIM(PS)_2_ Difluor (Rp/Sp). Downstream IL-1β (Figure 1D–F), IL-6 (Figure 1G–I), IL-10 (Figure 1J–L) and TNFα (Figure 1M–O) production was measured in supernatants, alongside cell viability (Figure 1P–R) and IFN-β production (Figure 1S). This initial screen was performed using a single clone of C5aR2 KO THP-1 cells (F7) with five technical replicates. C5aR2 KO caused a significant reduction in both Pam3CSK4 and LPS-induced IL-6 (Figure 1G,H), and LPS-induced TNFα (Figure 1N), and a significant yet variable increase in STING-induced IL-1β (Figure 1F). However, the most striking result from this screen was the enhanced IFN-β production in C5aR2 KO cells following STING agonism (Figure 1S), which warranted further investigation.

### 3.3. Characterising the Enhanced IFN-β Response to STING Agonism in C5aR2 KO THP-1 Cells

To further characterise the elevated IFN-β response observed in C5aR2 KO THP-1 cells upon STING agonism, the kinetics of IFN-β secretion in this context were assessed. These and all subsequent experiments were performed using three independent C5aR2 KO THP-1 cell line clones. The enhanced IFN-β response in C5aR2 KO cells was conserved between clones and correlated with increasing STING agonist concentration at a set time point of 6 h (Figure 2A). The time point of 6 h was selected to prevent the minor reduction in viability observed at 24 h post STING agonism (Figure 1R). At 6 h, only the top concentration of STING agonist showed a minor impact on viability (Figure 2B). C5aR2 KO THP-1 cells were then stimulated with 5 µg/mL STING agonist, selected to maximise the differential between wild-type and KO (Figure 2A) without impacting viability (Figure 2B), and IFN-β secretion was detected across a range of incubation times (Figure 2C). Compared to wild-type, C5aR2 KO THP-1 cells secreted higher levels of IFN-β at time points where IFN-β was detected (Figure 2C) and viability was retained until 24 h, where a small decrease was observed (Figure 2D). The enhanced IFN-β response was replicated with the cGAS agonist, G3-YSD (Figure 2E) and was blocked by antagonists of TBK1 (BX795; Figure 2F) and STING (H-151; Figure 2G). Taken together, these data indicate that C5aR2 KO markedly enhanced the IFN-β response to cGAS-STING pathway activation.

### 3.4. Assessing Expression and Phosphorylation of STING Pathway Proteins

Given the enhanced IFN-β response to STING agonism in C5aR2 KO THP-1 cells, we investigated the expression of proteins in the cGAS-STING pathway. Wild-type and C5aR2 KO THP-1 cells were stimulated with 5 µg/mL STING agonist for 6 h, and expression of total and phosphorylated STING and IRF3, and loading control β-actin, was assessed by semi-quantitative capillary electrophoresis (Figure 3A,B). Peak area data for total STING (Figure 3C), activated STING (Figure 3D), phospho-STING (Ser366) (Figure 3E), total IRF3 (Figure 3F) and phospho-IRF3 (Ser386) (Figure 3G) were plotted as a ratio to β-actin to determine relative expression levels. C5aR2 KO THP-1 cells had a slightly increased level of activated STING compared to the wild-type control, and an increased level of phosphorylated IRF3, indicating that C5aR2 KO increases signal transduction downstream of STING activation.

### 3.5. Confirmation of an Enhanced IFN-β Response to STING Agonism in Primary Human Macrophages with Reduced C5aR2

To explore C5aR2 on STING-induced IFN-β secretion in a more physiologically relevant system, primary human monocyte-derived macrophages (hMDMs) from five independent donors were edited using CRISPR-Cas9 targeted against C5aR2. These hMDMs were edited by CRISPR-Cas9 in the same fashion as above, except with a triple guide approach to increase editing efficiency [17,18], and without the monoclonal dilution phase, resulting in a polyclonal C5aR2-edited population. After 10 days post-editing to allow for protein turnover in the terminally differentiated cell model, Sanger sequencing was used to confirm editing efficiency in these cells (Appendix A), followed by stimulation with STING agonist for 6 h and cytokine detection in cell supernatants (Figure 4A–G, Appendix A). IFN-β in the supernatants was markedly enhanced in C5aR2 edited hMDMs (Figure 4A, Appendix A), albeit with variability between the primary donors and with two donors exhibiting a reduced response to stimulation. The three responsive donors were selected to investigate IL-1β, IL-6, IL-10, TNFα, IL-18 and GM-CSF secretion by Luminex, all of which were produced similarly between non-targeted controls (NTC) and C5aR2 edited hMDMs (Figure 4B–G). This data shows that the STING-mediated production of IFN-β was enhanced in C5aR2 edited primary hMDMs.

### 3.6. Assessing Transcriptomic Differences between Anaphylatoxin Receptor Knockout THP-1 Cell Lines

We have established that C5aR2 has a role in regulating IFN-β production downstream of STING pathway activation. To further investigate the downstream consequences of this relationship, we performed unbiased bulk RNASeq on STING-agonised wild-type and C5aR2 KO THP-1 cells. C5aR1 KO monoclonal THP-1 cell lines were also included to investigate the specific nature of the observation to C5aR2, and cells were also stimulated with C5a, the putative natural ligand for C5aR1 and C5aR2, to investigate the specific nature of STING agonism.

First, C5aR1 KO THP-1 cells were generated using CRISPR-Cas9 with a triple-guide approach, followed by monoclonal cell line generation and C5aR1 KO confirmation using Sanger sequencing (Appendix A), fluorescent microscopy (Appendix A) and flow cytometry (Appendix A). Wild-type, C5aR1 KO and C5aR2 KO THP-1 cells were then stimulated with C5a or STING agonist, followed by RNASeq. First, PCA was utilised to assess the variance associated with treatment and genotype in C5aR1 KO clones (Appendix A) and C5aR2 KO clones (Appendix A), which confirmed that STING agonism drove the majority of variance in each dataset, and that genotypes clustered, indicating that transcriptomic differences were primarily driven by genotype and treatment condition, as expected. Differentially regulated genes between all conditions were then represented in volcano plots (Appendix A), which revealed significant differentially regulated gene expression across all conditions. In C5a-treated THP-1 cells, C5aR1 KO abrogated the response seen in the wild-type, as expected. The response to STING agonism in C5aR2 KO THP-1 cells had higher levels of differentially regulated genes compared to wild-type cells and C5aR1 KO cells.

Further analysis was performed to elucidate specific genes and pathways that were regulated by C5aR2 KO, using three complementary approaches (Appendix A). First, GSEA was performed using all genes in an unbiased manner (Appendix A). In untreated C5aR2 KO cells compared to untreated wild-type cells, cytosolic DNA sensing pathways were significantly up-regulated, suggesting an enhancement of cytosolic antiviral surveillance at baseline in C5aR2 KO cells. Upon stimulation with STING agonist, C5aR2 KO cells retain their up-regulated cytosolic DNA sensing pathways and exhibit up-regulated JAK-STAT Signalling pathways compared to wild-type cells, which is indicative of an enhanced response to STING-induced IFN-β via downstream IFN α/β receptor activation. In contrast, these pathways were not up-regulated in C5aR1 KO cells treated with STING agonist (Appendix A). Pathways such as cell cycle and DNA replication were associated with both CRISPR-Cas9-edited cell lines, suggesting a potential CRISPR-induced transcriptomic signature at baseline.

To address this, and to investigate the treatment-specific responses within this analysis, differentially expressed genes (log2 fold change ≤−1 and ≥1, and *p* ≤ 0.05) were filtered to identify unique genotype-dependent, treatment-dependent differentially regulated genes (Figure 5A). These genes were analysed using STRING to construct an in silico interactome and highlight significantly regulated KEGG pathways (false discovery rate ≤ 0.01) represented within clustered genes (Figure 5B, Appendix A). This approach aimed to reduce the impact of responses generic to C5a and STING agonism, potential crosstalk between the anaphylatoxin receptors, and off-target effects of genome editing, in order to enrich for the most physiologically relevant differentially regulated genes and pathways. The JAK-STAT signalling pathway was significantly up-regulated, including a broad range of Type I IFNs and cytokines, confirming the antiviral phenotype reported by GSEA. A pathway analysis was then performed using PANTHER and GO biological process gene lists (Figure 5C), which yielded a similar result. In response to STING agonism, response to virus, cellular response to virus, cellular response to Type I interferon and Type I interferon signalling pathways were significantly up-regulated in C5aR2 KO cells compared to wild-type cells, in concordance with the previous analyses.

Genes in these pathways are potential interactors or indirect targets of C5aR2, therefore contributory genes were identified. A set of significantly regulated KEGG pathways was manually selected, based on prior knowledge of the innate immune response (JAK-STAT signalling, cytosolic DNA sensing, NLR signalling, RLR signalling, cytokine-cytokine receptor interaction and adipocytokine signalling). Significantly differentially regulated contributory genes were visualised amongst all differentially expressed genes in a volcano plot (Figure 5D). This approach highlighted bidirectional regulation of a broad range of genes across these pathways, including a variety of IFNs and cytokines, and provides a strong basis on which to continue research into the mechanisms underlying the function of C5aR2. Further investigation into these highlighted pathways and most differentially regulated genes is warranted in future studies.

## 4. Discussion

C5aR2 is the less understood of the two anaphylatoxin receptors, with some evidence suggesting a role in modulation of innate immune responses through negative regulation of C5aR1 signalling and PRR-induced cytokine secretion. Lack of specific, well-characterised tools to interrogate C5aR2 function have hampered furthering the understanding of its biology. The role of C5aR2 in disease models, albeit conflicting in directionality [9], suggests a possible role of C5aR2 in inflammatory disease which warranted our attempt to apply genetic editing as a tool to provide further insight into the function of C5aR2 as a regulator of inflammatory signalling in myeloid cells. We further elucidated the immunomodulatory effect of C5aR2 on PRR signalling in human macrophages and identified a role for C5aR2 in regulation of the IFN response to STING agonism. This finding adds to the limited field of C5aR2 research and opens up possibilities for further molecular and disease-specific investigations to expand much needed understanding of C5aR2 in pathology, and potential therapeutic opportunity.

### 4.1. C5aR2 Modulates PRR Signalling

C5aR2 has been implicated in immunomodulation in previous studies which investigated the C5aR2-dependent regulation of C5aR1 and β-arrestin, which are known regulators of the MAPK pathway [10,14,15]. Activation of C5aR2 with a peptide agonist has been shown to negatively regulate TLR4-induced IL-6 secretion [14], TLR3/TLR4/TLR7/Mincle/STING-induced IL-6 secretion and Dectin-2/Mincle/STING-induced TNFα secretion [16]. In vivo, pharmacological inhibition and genetic ablation of C5aR2 has shown conflicting results which may be due to differences in mouse strains, the tools used and the mechanistic basis of each disease model applied. Based on these published observations, we hypothesised that C5aR2 may have a role in regulating crosstalk during the innate immune response, linking the complement and PRR axes.

We observed that C5aR2 KO THP-1 cells exhibited alterations in TLR1/2, TLR4 and STING-induced cytokine secretion. TLR-mediated IL-6 and TNFα secretion was significantly down-regulated (Figure 1), suggesting C5aR2 could regulate these responses in wild-type cells. However, the directionality observed is opposite to Li et al. [16]. Li et al. used a C5a-based peptide agonist, whereas observations made in the present study were independent of C5a. In this study, we explored C5a stimulation of THP-1 cells in the presence and absence of C5aR1 and C5aR2, which yielded a modest transcriptional response (Appendix A). The relevance of C5a and the observed link to PRR signalling will be explored in future studies, along with understanding the cellular localisation of C5aR2, elucidation of potentially distinct intra- and extracellular functions, as well as cell-type and context-specific roles. Further investigation into the role of C5a in C5aR2-mediated immunomodulation is required to fully understand the function of C5aR2 across diverse immunological contexts.

### 4.2. C5aR2 KO Amplifies STING-Induced IFN-β Secretion and Modulates Antiviral Signalling Pathways

cGAS/STING signalling is a critical component of innate immune signalling and is particularly relevant to pathogen sensing and the immune response to nucleic acids. Given previous observations using pharmacological modulators, we hypothesised that C5aR2 may have a role in regulating PRR biology, in particular STING signalling (Figure 1S). Pharmacological modulation using G3-YSD, cAIM(PS)_2_ Difluor (Rp/Sp), H-151 and BX795 confirmed that the enhanced IFN-β signature was dependent on cGAS, STING and TBK1 (Figure 2), and C5aR2 KO THP-1 cells up-regulated expression of STING and IRF3 at the protein level (Figure 3).

The response to STING agonism described here is in concordance with the observations made by Li et al. [16]. C5aR2 KO amplified STING-induced IL-6 secretion, IFN-β secretion, and phosphorylation of STING and TBK1, suggesting that C5aR2 negatively regulates these pathways. The observation was corroborated using CRISPR-Cas9 editing of C5aR2 in primary hMDMs, which also resulted in enhanced IFN-β secretion upon STING agonism (Figure 4).

Transcriptomic analysis of STING-agonised C5aR2 KO THP-1 cells confirmed that antiviral sensing pathways are up-regulated in C5aR2 KO cells at baseline, and JAK-STAT signalling is up-regulated following STING agonism (Figure 5). JAK-STAT signalling occurs downstream of IFN-α/β receptor activation and upstream of interferon-stimulated gene expression, further implicating C5aR2 as a regulator of the IFN response. Alongside IFN-β, a wide array of cytokines, chemokines and the Type I and III IFN responses were amplified in C5aR2 KO THP-1 cells, identifying potential mechanistic contributors to the C5aR2 KO phenotype. These include Z-DNA-binding protein 1 (ZBP1), a cytosolic DNA-sensing protein; ciliary neurotrophic factor receptor (CNTFR), which is closely related to the IL-6 receptor; and Leptin (LEP), which is hormone derived from adipose tissue, where C5aR2 has been implicated as a regulator of immunometabolism [28]. Future studies could expand these observations from transcriptomic analyses into primary cell models or relevant in vivo models. Further investigation into these potential influencer genes will be critical for defining the molecular mechanisms underlying the C5aR2-dependent regulation of the cGAS-STING pathway and IFN-β.

Previous observations suggested a link between the anaphylatoxin receptors and PRR signalling. Here, we have confirmed a relationship between C5aR2 and cGAS-STING signalling, harnessing the specificity of genetic editing to characterise C5aR2 as a modulator of the innate immune response. There are various differences between this study and previously published studies, including the use of cell lines compared to primary cells, the presence or absence of C5a, and pharmacological versus genetic manipulation. Taken together, these observations positively contribute to our understanding of the impact of C5aR2 on innate immune signalling beyond the complement axis, providing rationale for further characterisation of the observed interactions.

### 4.3. Therapeutic Potential

As a sensor of cytosolic DNA, cGAS-STING signalling is highly relevant in infectious disease with several examples of the clinical efficacy of STING inhibitors [29]. Equally, C5aR2 has been implicated in the response to *Neisseria meningitidis* and *Listeria monocytogenes,* where C5aR2 KO was protective [30,31]. Similarly, C5aR1 has been implicated in viral pathologies, including COVID-19 and influenza-induced lung injury, with positive efficacy using anti-C5aR1 therapies [32,33,34]. The homology between C5aR1 and C5aR2, and new proposed link to cGAS-STING signalling, provide rationale for further investigations C5aR2 in other infection scenarios.

C5aR2 has also been implicated in inflammatory processes [9], including NLRP3 inflammasome regulation in macrophages [35] and regulation of neutrophilic inflammation in epidermolysis bullosa acquisita [36] and arthritis [37]. These data extend the evidence for C5aR2 in myeloid cell biology and disease, however, this is likely tissue and organ-specific as the consequences of C5aR2 interference are not consistent across disease contexts [8] (reviewed in [9]). Other IFN-related conditions, such as STING-associated vasculopathy with onset in infancy (SAVI) and tauopathies, including Alzheimer’s disease, both linked to dysregulated Type I IFN responses, warranting further research of C5aR2 in these contexts [38,39].

In oncology, the role of C5aR2 remains controversial with links to immune infiltration, macrophage polarisation, and regulation of tumourigenic pathways in breast cancer [7] and a protective association in a murine model of melanoma [40]. The cGAS-STING pathway is critical for anti-tumour immunity via the recognition of damage-associated molecular patterns (DAMPs), primarily tumour-derived DNA [41], therefore inhibition of C5aR2 may amplify the host innate immune response to drive tumour clearance. However, further context specific work is required here.

### 4.4. Proposals for Future Work

Here, we investigated the role of C5aR2 in the context of STING signalling completely independently of the putative ligand C5a. We propose that here, C5aR2 acts through an unknown interactor or independently of C5a to regulate STING signalling. Understanding the source and consequences of C5a on its putative receptors is critical to evaluate the relevance of C5aR1 and C5aR2 in in vivo and disease settings. C5aR2 may not require extracellular C5a and may instead be regulated by the complosome [38], as is observed for C5aR1 [39]. It is highly likely that C5aR2 and associated functions are partially dependent on subcellular location and accessibility to exogenous C5a and unknown interactors. To date, there is a lack of consistency on the subcellular location of human C5aR2 and further understanding here is critical to appreciate the potential relevance of these observed phenomena.

Aside from location, expanded interactome studies, such as tagged immunoprecipitation, would cast light on interaction partners and elucidate key molecular details of C5aR2 function. Key interactors could then be validated using pharmacological or genetic approaches to abrogate interactions and assess function, including the observed IFN-β amplification seen here. Additional validation of the observed phenotype using pharmacological approaches, such an intracellular delivery of an anti-C5aR2 blocking antibody or peptide agonists, would further validate the observed immunological response [14]. There is also scope to further characterise the role of C5aR2, and the consequential changes in IFN-β production, on functional macrophage biology such as chemotaxis, phagocytosis and cytokine/chemokine secretion, especially as the observed phenotype was consistent in primary macrophage models.

Additionally, there is an intriguing opportunity to probe further into the transcriptomic dataset which is publicly available (GSE245973). Further in-depth analysis of key pathways in response to C5a, or in C5aR1 KOs may yield further insights into an already limited field of complement biology, as well as understanding the baseline response to C5a and/or STING agonists is an exciting prospect for the C5aR1/2 area with therapeutic potential. Fundamentally, a deeper molecular characterisation of the C5aR2-STING relationship would be critical to fully appreciate the potential that we have highlighted here.

### 4.5. Conclusions

This study has highlighted a role for C5aR2 as a regulator of nucleic acid sensing and the IFN-β response in human macrophages. We have suggested a plethora of future studies to aid the expansion of the C5aR2 field in this new context and use this article to propose the initial link between C5aR2 and STING signalling. Investigating C5aR2 in the context of other nucleic acid sensing pathways may uncover whether C5aR2 is linked to nucleic acid sensing more broadly. However, understanding the interaction between C5aR2 and innate immune signalling pathways on a molecular level, beyond the macrophage as well as in in vivo or disease-relevant studies, will provide further insight into the relevance of C5aR2 as an immunomodulatory cog in the complement system.

## Figures and Tables

**Figure 1 cells-12-02707-f001:**
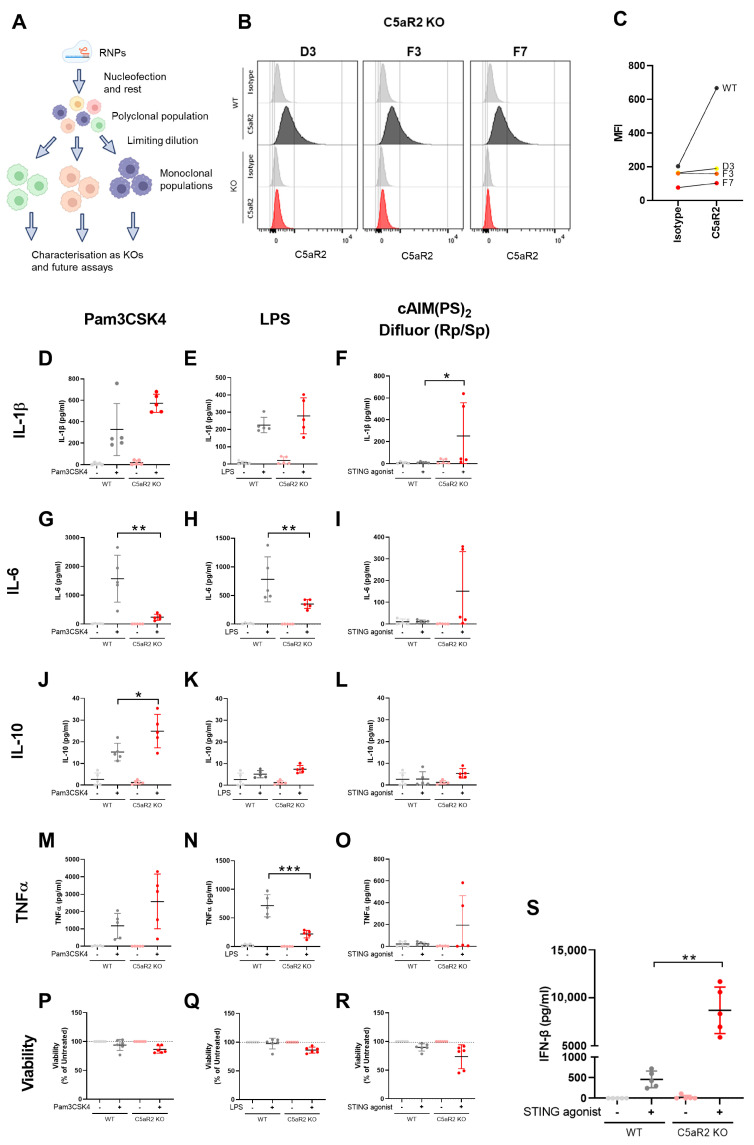
C5aR2 KO modulates PRR-induced cytokine secretion in THP-1 cells. (**A**) Schematic of CRISPR-Cas9 setup. (**B**) C5aR2 expression in PMA-differentiated wild-type and C5aR2 KO THP-1 cells analysed by flow cytometry to confirm protein knockout, and (**C**) MFI was plotted. (**D**–**S**) PMA-differentiated wild-type or C5aR2 KO (clone F7) THP-1 cells were stimulated with 10 μg/mL TLR1/2 ligand Pam3CSK4, 10 ng/mL TLR4 ligand LPS or 5 μg/mL STING ligand cAIM(PS)_2_ Difluor (Rp/Sp) for 24 h. Secretion of IL-1β, IL-6, IL-10 or TNFα was assessed by MSD U-Plex (**D**–**O**), and cells were assessed for viability by Cell Titre-Glo (**P**–**R**). (**S**) Secretion of IFN-β was assessed by ELISA. Mean ± SD, unpaired two-tailed Student’s *t* tests or Mann–Whitney U tests. N = 5 technical replicates across 3 separate experiments using a single C5aR2 KO clone. ns: *p* > 0.05; *: *p* ≤ 0.05; **: *p* ≤ 0.01; ***: *p* ≤ 0.001;. Grey; NTC, Red; C5aR2 edited, light; vehicle, dark; agonist.

**Figure 2 cells-12-02707-f002:**
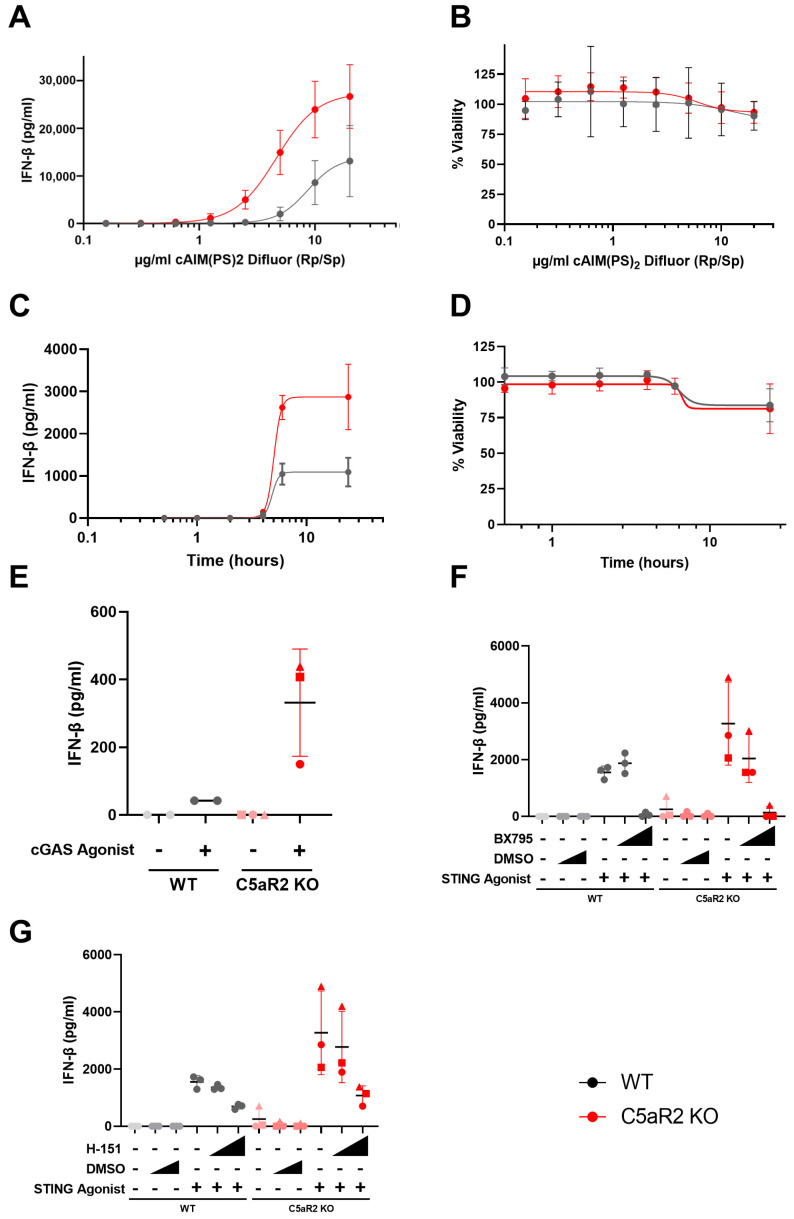
cGAS/STING-induced IFN-β secretion is amplified in C5aR2 KO THP-1 cells. (**A**,**B**) PMA-differentiated wild-type or C5aR2 KO (clones D3, F3, F7) THP-1 cells were stimulated with 0–20 µg/mL STING ligand cAIM(PS)_2_ Difluor (Rp/Sp) for 6 h and supernatant IFN-β (**A**) or viability (**B**) measured. (**C**,**D**) Cells were stimulated with 5 µg/mL STING ligand at time points up to 24 h with superntant IFN-β (**C**) and viability measured (**D**). (**E**) Cells were stimulated with 1 µg/mL cGAS ligand G3-YSD for 6 h and supernatant IFN-β measured. (**F**,**G**) Cells were pre-incubated with 100 nM or 1 µM TBK1 antagonist BX795 (**F**) or STING antagonist H-151 (**G**) for 30 min, then stimulated with 5 µg/mL STING ligand cAIM(PS)_2_ Difluor (Rp/Sp) for 6 h. Secretion of IFN-β was assessed by ELISA, and viability was assessed using Cell Titre-Glo. Mean ± SD, N = 2–3 wild-type replicates and C5aR2 KO clones noted by shapes; Circle: D3. Square: F3. Triangle: F7. Grey; NTC, Red; C5aR2 edited, light; vehicle, dark; agonist.

**Figure 3 cells-12-02707-f003:**
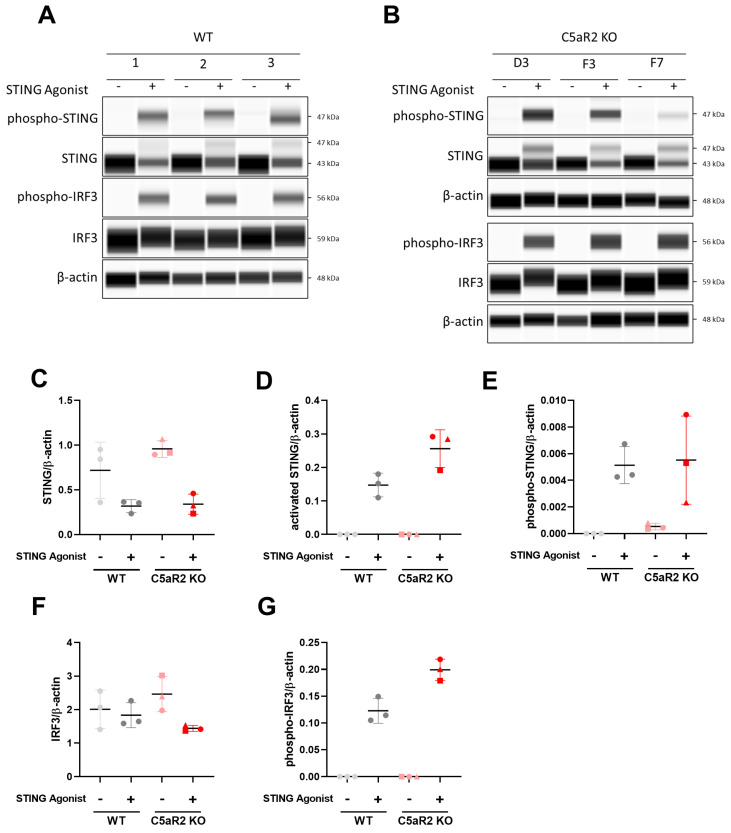
C5aR2 KO amplifies expression and phosphorylation of cGAS-STING pathway proteins. PMA-differentiated wild-type or C5aR2 KO THP-1 cells were stimulated with 5 μg/mL STING agonist cAIM(PS)_2_ Difluor (Rp/Sp) for 6 h. (**A**,**B**) Lysates were assessed for expression and phosphorylation of cGAS-STING pathway proteins STING and IRF3 by Peggy Sue. (**C**–**G**) Area under the curve for each protein was normalised to β-actin. Mean ± SD. N = 3 wild-type replicates and C5aR2 KO clones represented by shapes; Circle: D3. Square: F3. Triangle: F7. Grey; NTC, Red; C5aR2 edited, light; vehicle, dark; agonist.

**Figure 4 cells-12-02707-f004:**
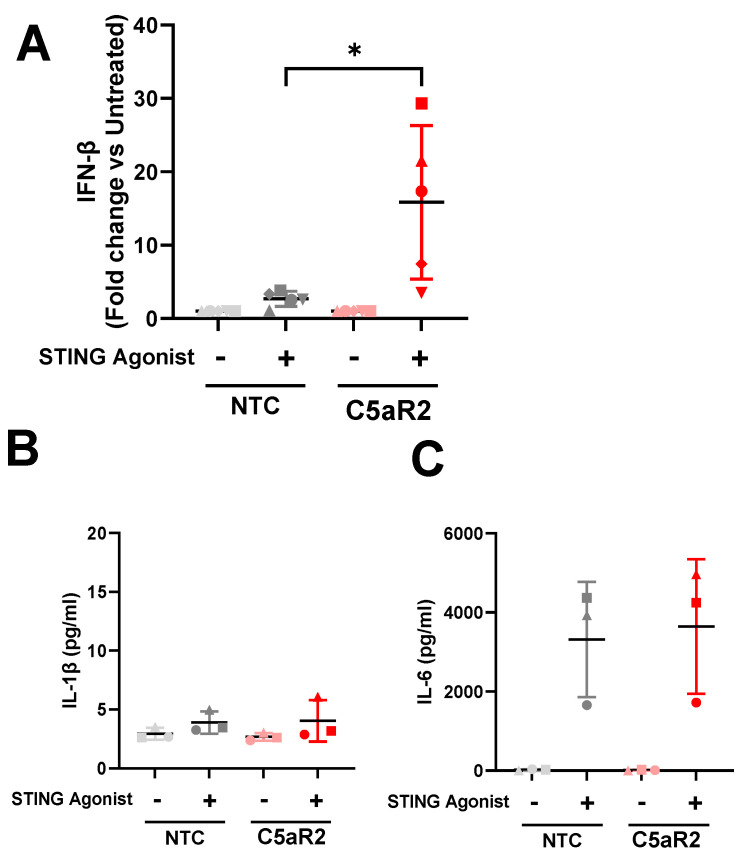
Reduction of C5aR2 amplifies STING-induced IFN-β secretion in primary human monocyte-derived macrophages. Non-targeted control (NTC) and C5aR2-edited (C5aR2) primary hMDMs were stimulated with 5 μg/mL STING agonist cAIM(PS)_2_ Difluor (Rp/Sp) for 6 h. Supernatants were assessed for (**A**) IFN-β by ELISA, or (**B**–**G**) IL-1β, IL-6, IL-10, TNFα, IL-18 and GM-CSF by Luminex. Mean ± SD, paired Student’s *t* test. (**A**) N = 5 independent donors, (**B**–**G**) N = 3 independent donors. Fold change between unstimulated and stimulated per genotype was plotted for primary IFN-β to account for donor-to-donor variation in response to agonist. Grey; NTC, Red; C5aR2 edited, light; vehicle, dark; agonist. *: *p* ≤ 0.05.

**Figure 5 cells-12-02707-f005:**
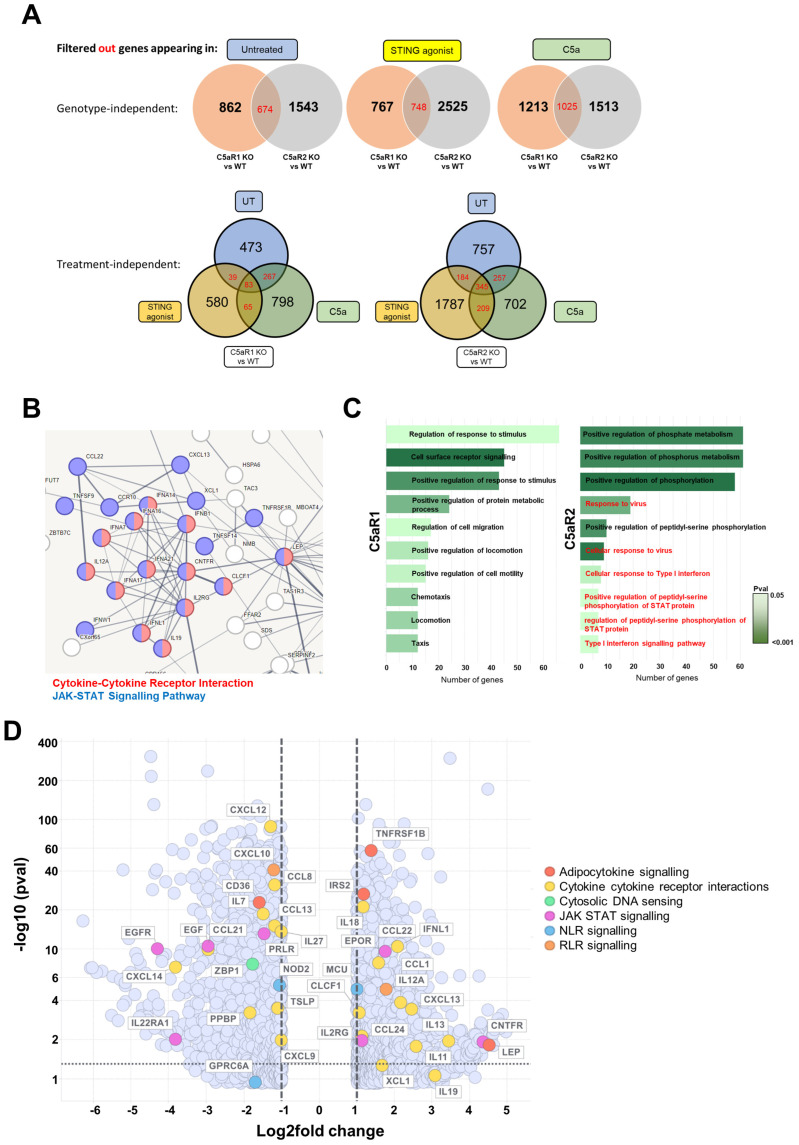
Pathway analysis of transcriptomic data reveals an enhanced antiviral phenotype in C5aR2 KO THP-1 cell lines compared to WT. (**A**) Unique, genotype-dependent, treatment-dependent significantly regulated genes (*p* ≤ 0.05, log_2_FC < ±1) were identified. (**B**) An unbiased cluster analysis and pathway analysis was performed using STRING, and the significantly enriched pathways were highlighted in red (Cytokine-Cytokine Receptor Interaction) or blue (JAK-STAT Signalling Pathway) (Appendix A). (**C**) An unbiased pathway analysis was performed using PANTHER and GO Biological Process gene sets. Red text highlights relevant anti-viral pathways (**D**) Significantly regulated KEGG gene sets from the GSEA were manually triaged, selecting for unique pathways for each condition, genotype-specific stimulation-independent pathways, or genotype-independent stimulation-independent pathways, filtering out stimulation-specific genotype-independent pathways, or pathways common to all conditions. Significantly regulated inflammatory pathways from the C5aR2 KO vs. wild-type comparison (C5a or STING agonist-treated) were manually curated. Selected genes from KEGG gene sets were marked in a volcano plot for STING agonist-treated C5aR2 KO cells vs. STING agonist-treated wild-type cells.

**Table 1 cells-12-02707-t001:** Guide sequences used in this publication.

Cells	Gene	Number	Sequence
THP-1	C5aR1	1	ATCAGGGGTGGTATAATTGA
		2	TTTTATCCACAGGGGTGTTG
		3	CTGCAAAGATGACCAAGGCC
	C5aR2	1	CCGCTGAACCGTAGACCACC
hMDM	NTC	1	GCACTACCAGAGCTAACTCA
	C5aR2	1	ATTCTGTCAGCTACGAGTAT
		2	GGCCATCGACCCGCTGCGCG
		3	GGGGGTGCCGGGCAATGCCA

**Table 2 cells-12-02707-t002:** Primer sequences used in this publication.

Cells	Gene	Primer	Sequence
THP-1	C5aR1	Forward	GCAGGAGAGGAAGTCGGCTA
		Reverse	AGAAAAAGCCACACAGGGGA
	C5aR2	Forward	CATGGAGTTTCCTCCTCTGAGT
		Reverse	GCCAAAAAGAAACCGGATGG
hMDM	C5aR2	Forward	AAGCACTGGAGTCCTTATGACG
		Reverse	ACAAACAGCACAGCAAATCCG

## Data Availability

RNASeq analysis dataset is available through GEO using ID GSE245973.

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
