# Peer review of "C5aR2 Regulates STING-Mediated Interferon Beta Production in Human Macrophages"

_cells, 2023, doi:10.3390/cells12232707_

Round 1
Reviewer 1 Report
Comments and Suggestions for Authors
In this study, the authors claim that loss of C5aR2 enhances the cGAS-STING pathway using human macrophage cells deficient in C5aR2. The experiments are generally well done and the experimental results are reasonable. However, the molecular mechanism by which C5aR2 deficiency enhances the cGAS-STING pathway is not clear. For example, IRF3 activation appears to be unaffected (in Figure 3; indeed, this needs to be assessed quantitatively), but the STING activation is affected; how C5aR2 signaling regulates the cGAS-STING pathway is a question that requires additional detailed pathway analysis. The authors are encouraged to perform additional detailed molecular cascade analyses.
Minor points.
The text in the figures is very small and difficult to read and needs improvement throughout.
Author Response
In this study, the authors claim that loss of C5aR2 enhances the cGAS-STING pathway using human macrophage cells deficient in C5aR2. The experiments are generally well done and the experimental results are reasonable. However, the molecular mechanism by which C5aR2 deficiency enhances the cGAS-STING pathway is not clear. For example, IRF3 activation appears to be unaffected (in Figure 3; indeed, this needs to be assessed quantitatively), but the STING activation is affected; how C5aR2 signaling regulates the cGAS-STING pathway is a question that requires additional detailed pathway analysis. The authors are encouraged to perform additional detailed molecular cascade analyses.
We would like to thank Reviewer 1 for their comments and suggestions. We agree the molecular mechanism is indeed the next step to deciphering the importance of the relationship between C5aR2 and nucleic acid signaling. We chose Cells due to their acceptance of Research Proposals, in which we felt this work fit well. We have added comments in the introduction and discussion to highlight the proposal flavor of this work, including a substantial section in the discussion proposing future experiments and areas of expansion, and we hope that by elucidating this link we can kickstart the limited C5aR2 field to investigate exciting prospects of C5aR2 in controlling inflammation.
Minor points.
The text in the figures is very small and difficult to read and needs improvement throughout.
Thank you for highlighting this. We have endeavoured to increase the text size where possible in all figures, including supplementary. In addition, we have moved Fig 5A, B to Supplementary Fig 5 to improve the text size for Figure 5, which we now appreciate was too small to read properly in the Cells formatting.
Reviewer 2 Report
Comments and Suggestions for Authors
In the manuscript entitled “C5aR2 Regulates STING-Mediated Interferon Beta Production in Human Macrophages”, authors analyzed molecular mechanism of C5aR2 in the context of PRR stimulation. In experiments, macrophage models lacking C5aR2 were generated using CRISPR-Cas9. Results indicated a significant increase in cGAS and STING-induced IFN-β secretion in C5aR2 THP-1 KO cells and C5aR2-edited primary human monocyte-derived macrophages. Moreover, expression of STING and IRF3 was increased in C5aR2 KO cell lines, suggesting that C5aR2 regulates the IFN-β response to cGAS-STING pathway activation.
Based on the manuscript, I have the following comments and questions:
1. How many leukopaks were used for the experiments?
2. In section 4.3 of the discussion there is written a following sentence: „This study has elucidated a role for C5aR2 as a regulator of nucleic acid sensing and the Type I IFN response in human macrophages.” However, there is no information in the Results section about STING-mediated IFN-α production? Can you clarify/add information about Type I IFN signalling?
3. Does STING agonism have any effect on IFN-β-mediated macrophage polarisation?
4. Have you assessed the functional properties of generated primary human monocyte-derived macrophages edited with C5aR2, such as phagocytosis and antigen presentation?
5. In the Future Perspectives section, you wrote about a possible important role in the treatment of viral infections. Do you see any prospects in the treatment of autoimmune diseases or cancer?
Author Response
In the manuscript entitled “C5aR2 Regulates STING-Mediated Interferon Beta Production in Human Macrophages”, authors analyzed molecular mechanism of C5aR2 in the context of PRR stimulation. In experiments, macrophage models lacking C5aR2 were generated using CRISPR-Cas9. Results indicated a significant increase in cGAS and STING-induced IFN-β secretion in C5aR2 THP-1 KO cells and C5aR2-edited primary human monocyte-derived macrophages. Moreover, expression of STING and IRF3 was increased in C5aR2 KO cell lines, suggesting that C5aR2 regulates the IFN-β response to cGAS-STING pathway activation.
Based on the manuscript, I have the following comments and questions:
- How many leukopaks were used for the experiments?
One individual leukopak were used for each donor, with five individual donors in total. Unfortunately, due to genetic editing consents we are only able to use a small portion of these samples for these experiments.
- In section 4.3 of the discussion there is written a following sentence: „This study has elucidated a role for C5aR2 as a regulator of nucleic acid sensing and the Type I IFN response in human macrophages.” However, there is no information in the Results section about STING-mediated IFN-α production? Can you clarify/add information about Type I IFN signalling?
Thank you for this suggestion. We initially referred to the Type I IFN pathways that were significantly up-regulated in the RNAseq experiment. Given that a macrophage cell line model was used, and that in primary cells, it is suggested that monocytes rather than macrophages are the major source of IFN-α (Congy-Jolivet et al., 2022, PMID 35561451), we felt it was appropriate to limit the comment to IFN-β, which we specifically focused on throughout this study. Future work using primary models could more thoroughly characterize the IFN response and indeed it would be interesting to look at other Type I IFNs in relevant, or other, settings.
- Does STING agonism have any effect on IFN-β-mediated macrophage polarisation?
Thank you for this question. We would suspect that STING agonism during IFN-β mediated polarisation would complement exogenous IFN-β through production of further IFN-β and subsequent autocrine and paracrine signalling. However, we are particularly interested in this part of this narrative and hope to follow up in time with a more detailed molecular understanding of the pathway as well as functional studies, including similar experiments to that you address here.
- Have you assessed the functional properties of generated primary human monocyte-derived macrophages edited with C5aR2, such as phagocytosis and antigen presentation?
Thank you for this excellent suggestion and something we plan to follow up with a more detailed functional based publication, as mentioned above. For this study in question, the primary cells we had access to with suitable consents for genetic editing were extremely limited in number, therefore limiting the number of experiments we were able to perform. We agree with the reviewer that this is an exciting proposition. Future work could be done to assess the effect of C5aR2 on classical macrophage function such as phagocytosis and antigen presentation, particularly in the presence and absence of STING agonism to determine if the increase in DNA sensing and IFN response also drives an increase in baseline macrophage function, or if these are separate phenomena. In order to aid the field, we have added an extensive section in the discussion highlighting potential future work of which we have included this suggestion.
- In the Future Perspectives section, you wrote about a possible important role in the treatment of viral infections. Do you see any prospects in the treatment of autoimmune diseases or cancer?
We agree with the reviewer that there is scope for broader speculation for future disease perspectives. We have identified a fundamental innate immune sensing mechanism that has relevance across a wide range of infectious and inflammatory disease. We have expanded the section in the discussion to include more reference to cancer, neurodegenerative disease, infectious disease and inflammatory disease
Reviewer 3 Report
Comments and Suggestions for Authors
Overall this manuscript in good shape. My concern is mainly about the unclarified molecular mechanism of C5aR2-IFN-β regulation.
1. Following the report that C5aR2 agonists downregulated production of cytokines including those mediated by STING (ref), authors decided to study the relevance of C5aR2 in PRR signaling and noticed C5aR2 depletion primed macrophage for production of cytokines downstream of STING (IL-1β and IFN-β). Fig 3 showed that C5aR2 depletion facilitated IRF3 phosphorylation but not STING phosphorylation. After STING activation, pro-inflammatory cytokines is mediated by NF-κB while IFN-β production is mediated by TBK1-IRF3 axis. Fig 4 showed induction of IFN-β but not pro-inflammatory cytokines in human macrophages indicating C5aR2 regulated STING-mediated IFN-β production through IRF3. Therefore, how C5aR2 regulates IRF3 requires investigation. Otherwise, this study has limited novelty and significance. Fig 5 showed higher activation of JAK-STAT signaling upon STING activation in C5aR2 depleted cells. There is crosstalk between IFN-β-JAK-STAT and TBK1-IRF3. Author may test if JAK-STAT contribute to enhanced IRF3 activity. Make good use of the RNA-seq data.
2. Authors used p-IRF3 (S386) antibody to evaluate IRF3 activaiton. IRF3 phosphorylation at S396 mainly contribute to IRF3 activity while S386 strengthen S396’s effect. Sometimes S396 phosphorylation detection is hard. In addition to S386 phosphorylation detection, authors should provide more evidence such as IRF3 dimerization or nuclear translocation to convince IRF3 activity control by C5aR2.
3. Authors wrote in the abstract that “expression of STING and IRF3 was increased in C5aR2 KO cell lines”. However, there were no significant difference of STING or IRF3 between WT and KO cells in Fig 3C and F.
Author Response
Overall this manuscript in good shape. My concern is mainly about the unclarified molecular mechanism of C5aR2-IFN-β regulation.
- Following the report that C5aR2 agonists downregulated production of cytokines including those mediated by STING (ref), authors decided to study the relevance of C5aR2 in PRR signaling and noticed C5aR2 depletion primed macrophage for production of cytokines downstream of STING (IL-1β and IFN-β). Fig 3 showed that C5aR2 depletion facilitated IRF3 phosphorylation but not STING phosphorylation. After STING activation, pro-inflammatory cytokines is mediated by NF-κB while IFN-β production is mediated by TBK1-IRF3 axis. Fig 4 showed induction of IFN-β but not pro-inflammatory cytokines in human macrophages indicating C5aR2 regulated STING-mediated IFN-β production through IRF3. Therefore, how C5aR2 regulates IRF3 requires investigation. Otherwise, this study has limited novelty and significance. Fig 5 showed higher activation of JAK-STAT signaling upon STING activation in C5aR2 depleted cells. There is crosstalk between IFN-β-JAK-STAT and TBK1-IRF3. Author may test if JAK-STAT contribute to enhanced IRF3 activity. Make good use of the RNA-seq data.
We would like to thank Reviewer 3 for their comments and suggestions. We agree that it is likely that STING agonism in C5aR2 edited primary macrophages drives enhanced IFN-β via IRF3, as seen in Figure 4, and we agree that furthering our understanding of the IRF3-C5aR2 context is important. We have submitted this work to Cells under the Research Proposals context, reflecting the somewhat preliminary and nature of the work. We fully intended for this work to reach the limited C5aR2 field to aid researchers in further research in this important and potentially exciting area. We do intend to follow up with further work characterizing the functional and molecular details associated with this work. Included in this would be detailed molecular characterisation of the interaction as well as functional effects of C5aR2 KO in primary macrophages. Although we did see changes with other PRR agonists in Fig 1, this was performed in Thp1s and not primary macrophages, and our observations of other cytokines in primary macrophages did not repeat from Thp1s, hence our focus on IFN-β. Repeating these observations will aid in deciphering the specificity of C5aR2 interactions to PRR agonism.
To ensure we have pitched the publication in the right tone, we have added a substantial future work and relevance section into the discussion as well as changing the language throughout to reflect the proposal nature of the article.
- Authors used p-IRF3 (S386) antibody to evaluate IRF3 activaiton. IRF3 phosphorylation at S396 mainly contribute to IRF3 activity while S386 strengthen S396’s effect. Sometimes S396 phosphorylation detection is hard. In addition to S386 phosphorylation detection, authors should provide more evidence such as IRF3 dimerization or nuclear translocation to convince IRF3 activity control by C5aR2.
We agree with the reviewer that further investigation of IRF3 in this context is warranted. This antibody was preferentially selected as the only that was compatible with our capillary electrophoresis setup. As Reviewer 3 rightly points out pS396 is difficult and was not detected using our system. We have addressed our intention to investigate molecular mechanisms more thoroughly in future studies in the responses above, of which further understanding of IRF3 would certainly be warranted. This could certainly expand to other phosphorylation sites.
- Authors wrote in the abstract that “expression of STING and IRF3 was increased in C5aR2 KO cell lines”. However, there were no significant difference of STING or IRF3 between WT and KO cells in Fig 3C and F.
Thank you for highlighting this. We have altered the abstract accordingly to reflect the trend of increased expression, albeit not significant. We have refrained from performing statistics on data associated with figure 3 as this technique is only semi-quantitative.
Round 2
Reviewer 2 Report
Comments and Suggestions for Authors
The authors sufficiently addressed most of my questions and comments.
Reviewer 3 Report
Comments and Suggestions for Authors
Authors' reply is acceptable.